# An Adaptive Hybrid Model for Wind Power Prediction Based on the IVMD-FE-Ad-Informer

**DOI:** 10.3390/e25040647

**Published:** 2023-04-12

**Authors:** Yuqian Tian, Dazhi Wang, Guolin Zhou, Jiaxing Wang, Shuming Zhao, Yongliang Ni

**Affiliations:** 1College of Information Science and Engineering, Northeastern University, Shenyang 110819, China; 2China North Vehicle Research Institute, Beijing 100072, China

**Keywords:** wind power prediction, improved variational mode decomposition, fuzzy entropy, adaptive loss function, Informer

## Abstract

Accurate wind power prediction can increase the utilization rate of wind power generation and maintain the stability of the power system. At present, a large number of wind power prediction studies are based on the mean square error (MSE) loss function, which generates many errors when predicting original data with random fluctuation and non-stationarity. Therefore, a hybrid model for wind power prediction named IVMD-FE-Ad-Informer, which is based on Informer with an adaptive loss function and combines improved variational mode decomposition (IVMD) and fuzzy entropy (FE), is proposed. Firstly, the original data are decomposed into *K* subsequences by IVMD, which possess distinct frequency domain characteristics. Secondly, the sub-series are reconstructed into new elements using FE. Then, the adaptive and robust Ad-Informer model predicts new elements and the predicted values of each element are superimposed to obtain the final results of wind power. Finally, the model is analyzed and evaluated on two real datasets collected from wind farms in China and Spain. The results demonstrate that the proposed model is superior to other models in the performance and accuracy on different datasets, and this model can effectively meet the demand for actual wind power prediction.

## 1. Introduction

The global energy-shortage problem is becoming more and more serious, and it is essential to accelerate the pace of energy structure transformation based on the increasing proportion of renewable energy. Wind power, as an economical and environmentally friendly emerging renewable energy source, has been vigorously developed by various countries, and its application prospects are promising [1,2]. However, with the random fluctuation of wind power, it has strong uncontrollability, resulting in a decrease in the dispatching efficiency of the power grid and an imbalance between energy supply and demand [3]. Therefore, achieving high-accuracy and high-reliability prediction of wind power in practical applications can minimize energy loss and make the power grid operate more stably and safely.

In recent years, a large number of scholars have studied wind power prediction models, which can be mainly divided into physical models [4], statistical models [5], artificial intelligence (AI) models [6], and hybrid models [7]. The physical models are based on the method of fluid mechanics, which uses numerical weather prediction data to calculate the wind turbine output curve and then calculate wind power from it [8]. However, the fluid mechanics method has the disadvantages of the high complexity of building the model and massive computational cost. The statistical models are based on the mapping relationship between historical data and future data [9,10]. Rajagopalan et al. [11] proposed an autoregressive moving average (ARMA) model for ultra-short-term wind power forecasting and achieved superior results. The autoregressive integrated moving average (ARIMA) model is a widely used statistical model that is based on ARMA with the addition of difference operation [12,13]. However, this method requires a large-scale dataset, making it difficult to mine the nonlinear relationship of complex data.

AI models are current technical trends and are widely used in the field of large-scale and multi-dimensional data prediction [14]. AI models are mainly divided into machine-learning models and deep-learning models [15]. For example, an echo state network (ESN) [16] was applied to wind speed forecasting and improved the prediction performance. Khan et al. [17] used the Naive Bayes Tree (NB) to extract the probabilities of each feature of wind power, successfully predicting wind power values from hours to years. Machine-learning methods are based on rigorous mathematical theories that enable rapid computation in high-dimensional spaces. Owing to weak generalization ability, machine-learning methods are prone to overfitting, and it is difficult to achieve good prediction effects. In contrast, deep-learning models unify feature-learning tasks and prediction tasks into one model, making them more suitable than shallow machine-learning models to solve wind power prediction problems in complicated uncertainty scenarios [18]. Tian et al. [19] used a model based on the attention mechanism and demonstrated its efficacy in wind power prediction. Liu et al. [20] presented a novel deep convolutional neural network (CNN) capable of automatically extracting hidden information from multi-dimensional data and efficiently implementing multi-step prediction. Hu et al. [21] applied a model integrated with a deep-learning framework and basic ESN network for energy prediction, which enhanced the model’s memory capacity with a stacked hierarchy of reservoirs. Although these methods have achieved some success in wind power prediction, the fact that a single model cannot fully exploit the time series information leads to limited prediction performance [22].

Hybrid models are created by multiple intelligent algorithms or prediction models that combine the advantages of different models to achieve an improvement in prediction accuracy. Hybrid prediction models consist of combined multiple models and stacking models based on data processing [23,24]. Chen et al. [25] designed a weighted combination prediction composed of six long short-term memory networks (LSTM), and its prediction effect is better than that of a single prediction model. Xiong et al. [26] proposed a multi-scale hybrid prediction model that combines attention mechanism, CNN, and LSTM to adequately capture the high-dimensional features in wind farm data. Zheng et al. [27] established a hybrid model combining bidirectional long-short-term memory (Bi-LSTM) and CNN, which adopted a unique feature extraction method of space and then time. Although the combined prediction method of multiple models exhibits high prediction accuracy, it suffers low computational efficiency and narrow application scenarios [28]. Considering the nonlinear implicit relationship in the time series of wind power data, the stacking model based on data processing is proposed to improve the prediction accuracy by mining deep features through data decomposition. For example, Wu et al. [29] proposed a multi-step prediction method using variational modal decomposition (VMD) and chain ESN, which achieved multi-steps prediction at multiple time scales. Yang et al. [30] employed the VMD method to decompose the wind speed data, which was then utilized as input for an optimized LSTM network to perform predictions. Ren et al. [31] proposed a hybrid model of empirical-mode decomposition (EMD) and support-vector regression (SVR) for wind power prediction. Lv et al. [32] decomposed wind speed data into 3-dimensional input features using singular spectrum analysis (SSA) and fed them into a convolutional long-short-term memory (ConvLSTM) network, which effectively enhanced the local correlation between multivariate data. Khazaei and Ehsan [33] used prediction methods combining wavelet transform (WT) decomposition with the AI model, and the results showed that the model has high accuracy. Hybrid models based on data processing have a simple structure and strong feasibility, but the accuracy of the prediction greatly depends on the effectiveness of data decomposition. Over-decomposition of data can result in redundant components and reduce the efficiency of calculation, while the insufficient decomposition of data can lead to mode mixing, which fails to meet the needs of high-precision prediction [34].

Most prediction models use mean square error (MSE) as a loss function, which needs to meet the condition that prediction errors obey a Gaussian distribution. However, the use of MSE as a loss function in models that are insensitive to outliers in wind power data with high randomness may result in large errors [35]. In response to this issue, some researchers have improved the loss function of the model to minimize the impact of errors on the prediction results. Hu et al. [36] proposed a loss function without fixed distribution, which effectively solves the problem of prediction-gradient descent at wind power intervals. Duan et al. [35] designed a loss function with non-Gaussian distributed errors and combined it with an LSTM model to predict wind power. The loss function is significantly important in the wind power prediction process as it determines the training direction and accuracy of the model [37]. Although these improved loss functions have positive effects on wind power prediction, the models still exhibit limitations in terms of their adaptability and robustness.

Based on the above analysis, a robust and adaptive IVMD-FE-Ad-Informer hybrid model for wind power prediction is proposed in this paper, which aims to improve the precision of data decomposition and the predictive performance of non-stationary wind power data. The main contributions of this paper are outlined as follows: (1) Considering the difficulty in selecting the number of VMD, the IVMD algorithm improved by the maximum information coefficient (MIC) decomposes the original wind power data into *K* optimal sub-series, which effectively reduces the difficulty of wind power prediction. (2) fuzzy entropy (FE) is used to reconstruct sub-series into new elements of similar complexity together, alleviating the burden of model operation. (3) An adaptive loss function is innovatively introduced into the Informer network to solve the problem of traditional MSE’s insensitivity to randomly fluctuating wind power data. This novel model can reduce the impact of outliers in non-smooth wind power data. (4) Ablation experiments and comparative experiments are performed on datasets collected from both different wind farms to verify the effectiveness and stability of the model. The prediction results show that the proposed model framework is reasonable, and it exhibits significantly better prediction performance and accuracy compared to other models.

The specific contents of this paper are as follows: Section 2 specifically describes the basic methodologies of hybrid models; Section 3 presents the construction framework and evaluation indicators of the IVMD-FE-Ad-Informer model; Section 4 constructs four experiments to verify the accuracy and validity of the proposed model; and finally, this paper is summarized in Section 5.

## 2. Methodologies

### 2.1. Variational Mode Decomposition

VMD [38] is a commonly used data decomposition method that converts wind power sequences from the time domain to the frequency domain and subsequently decomposes them into *K* intrinsic mode functions (IMFs). Firstly, build the variational constraint equation:(1)$$\left\{ \begin{array}{l} \min\left\{\sum_{K=1}^{K}{\left\|\partial_{t}\left[\left(\delta (t)+\frac{i}{\pi t}\right)*u_{K}(t)\right]e^{-iw_{K}t}\right\|}_{2}^{2}\right\}\\ \;s.t.\;\sum_{K=1}^{K}u_{K}=x(t) \end{array}\right.$$
where ∗ is the convolution calculation symbol, x(t) is the wind power sequence, $w_{K}$ and $u_{K}$ are the central frequency and band components of the *k*th IMF value, δ(t) is the impulse function, and $\partial_{t}$ is used to denote the derivative of the function.

To simplify the variational constraint equation to a simple unconstrained problem, the Lagrange function λ(t) and the penalty factor α are introduced:(2)$$L\left(\left\{u_{K}\right\},\left\{w_{K}\right\},\lambda \right)=\alpha \sum_{K=1}^{K}{\left\|\partial_{t}\left[\left(\delta (t)+\frac{i}{\pi t}\right)*u_{K}(t)\right]e^{-iw_{K}t}\right\|}_{2}^{2}+{\left\|x(t)-\sum_{K=1}^{K}u_{K}(t)\right\|}_{2}^{2}+\langle \lambda (t),x(t)-\sum_{K=1}^{K}u_{K}(t)\rangle$$

Then the optimal solution of the unconstrained problem is solved using the alternating direction method of multiplication with the following iterative procedure:(3)$${\hat{u}}_{K}^{n+1}(w)=\frac{\hat{x}(w)-\sum_{i\neq K}{\hat{u}}_{j}(w)+(\hat{\lambda }(w)/2)}{1+2\alpha {\left(w-w_{K}\right)}^{2}}$$
(4)$$w_{K}^{n+1}=\frac{\int_{0}^{\infty }w{\left|{\hat{u}}_{K}^{n+1}(w)\right|}^{2}dw}{\int_{0}^{\infty }{\left|{\hat{u}}_{K}^{n+1}(w)\right|}^{2}dw}$$

Finally, after applying the above process, the original wind power series is decomposed into the *K* sub-series.

### 2.2. Fuzzy Entropy

Fuzzy entropy (FE) [39] is a dynamical method for analyzing the complexity of time series. The FE value changes smoothly with changes in the set parameters, which makes it more robust to noise and more resistant to interference. Firstly, for time series with the length of *n*, the FE algorithm is introduced into the fuzzy membership function, and the specific formula is as follows:(5)$$D(x)=\exp\left[-\ln(2){\left(\frac{x}{r}\right)}^{2}\right]$$
where *r* is the similarity tolerance, *x* = $d_{ij}^{m}$, $d_{ij}^{m}$ is the distance between vectors that reconstruct the time series into m-dimensional phase space, and *i*, *j* = 1,2..., *n* – *m* + 1, *i* ≠ *j*.

Averaging over each *i* in $D_{ij}^{m}$ yields, the average similarity function is as follows:(6)$$\phi^{m}(r)=\frac{1}{N-m+1}\sum_{i=1}^{N-m+1}\left(\frac{1}{N-m}\sum_{j=1,j\neq i}^{N-m+1}D_{ij}^{m}\right)$$

Therefore, the FE expression is as follows:(7)$$\mathrm{FuzzyEn}(m,r,n)=\ln\phi^{m}(r)-\ln\phi^{m+1}(r)$$

### 2.3. Informer

The Informer network is a variant of the Transformer that effectively addresses the long-sequence prediction problem [40]. Improvements of the Informer include: using a probsparse self-attention mechanism to reduce the complexity of matrix computation; introducing a self-attention distillation mechanism to extract the main features of time series, which effectively reduces memory usage; using a decoder to directly output the predicted values generatively to achieve the purpose of long-series prediction. The structure of the Informer model is shown in Figure 1.

The traditional self-attention mechanism consists of query, key, and value, and the expression is as follows:(8)fA(Q,K,V)=softmax(QK⊤d)V
where QϵℝLQ×d, KϵℝLK×d, VϵℝLV×d, *d* is the input dimension.

As the matrix multiplication involved in Equation (8) is computationally huge, the probsparse self-attention mechanism is introduced to select the important elements in Q to calculate the attention values.
(9)$$f_{A}(Q,K,V)=\mathrm{softmax}\left(\frac{\bar{Q}K^{⊤}}{\sqrt{d}}\right)V$$
where $\bar{Q}$ is obtained through probabilistic sparsity of Q and controlled by a constant sampling factor c and the number of $\bar{Q}$ is $c*\ln L_{K}$.

Therefore, the similarity and importance between query and key are measured by Kullback–Leibler divergence, as follows:(10)$$k(q_{i},k_{j})=\ln\sum_{l=1}^{L_{K}}e^{\frac{q_{i}k_{j}^{⊤}}{\sqrt{d}}}-\frac{1}{L_{K}}\sum_{j=1}^{L_{K}}\frac{q_{i}k_{j}^{⊤}}{\sqrt{d}}-\ln L_{K}$$
where the relevance of $q_{i}$ to $k_{j}$ is proportional to the magnitude of $k(q_{i},k_{j})$. If $p(k_{j}|q_{i})$ is close to a uniform distribution, i.e., $p(k_{j}|q_{i})=1/L_{K}$, indicating that $q_{i}$ has the same similarity to all $k_{j}$, then $q_{i}$ is deemed as a redundant vector and can be dropped.

Based on this, the sparsity evaluation formula that defines the *i*-th query is:(11)$$M\left(q_{i},K\right)=\ln\sum_{l=1}^{L_{K}}e^{\frac{q_{i}k_{l}^{⊤}}{\sqrt{d}}}-\frac{1}{L_{K}}\sum_{j=1}^{L_{K}}\frac{q_{i}k_{j}^{⊤}}{\sqrt{d}}$$

The self-attentive distillation mechanism is introduced in the encoder. The width of the feature map is reduced to half its length after the distillation layer, which can reduce the overall memory usage and effectively solve the problem of long input. The concrete representation is as follows:(12)$$X_{j+1}^{t}=f_{\mathrm{MP}}\;\left(\mathrm{ELU}\left(f_{\mathrm{Conv}}\left({\left[X_{j}^{t}\right]}_{AB}\right)\right)\right)$$
where *f*_MP_ represents the maximum pooling layer function, *f*_Conv_ denotes the convolutional layer function, and [·]*_AB_* is the attention unit.

The input of the decoder uses the time shield technique, and its input vector is as follows:(13)Xdec in =fConcat (Xtoken ,X0)∈ℝ(Ltoken +Lp)×dmodel 
where Xtoken ∈ℝLtoken ×dmodel  is the input start token, Ltoken is the length of the start token, X0∈ℝLp×dmodel  is the 0-value matrix, and Lp is the length of the part to be predicted.

### 2.4. Adaptive Loss Function

The adaptive loss function [41] obtains a generalized loss function by introducing robustness as a continuous parameter. During the training process, the adaptive loss function automatically adjusts the robustness parameters around the minimization loss algorithm, thereby enhancing the prediction accuracy. The generalized loss function formula is as follows:(14)$$f(z,\beta ,c)=\frac{\left|\beta -2\right|}{\beta }\left({\left(\frac{{\left(z/c\right)}^{2}}{\left|\beta -2\right|}+1\right)}^{\beta /2}-1\right)$$
where z is the difference between the true value and the predicted value, c>0 serves as a scale factor that controls the curvature of the quadratic function at x = 0, and β is a variable parameter that controls the robustness.

By analyzing Equation (14), the adaptive loss function changes with the change of β. For different β, the adaptive loss function formula is as follows:(15)$$L(z,\beta ,c)=\{\begin{array}{ll} \frac{1}{2}{\left(z/c\right)}^{2} & \;\mathrm{if}\;\beta =2\\ \log\left(\frac{1}{2}{\left(z/c\right)}^{2}+1\right) & \;\mathrm{if}\;\beta =0\\ \sqrt{{\left(z/c\right)}^{2}+1}-1 & \;\mathrm{if}\;\beta =1\\ 1-\exp\left(-\frac{1}{2}{\left(z/c\right)}^{2}\right) & \;\mathrm{if}\;\beta =-\infty \\ \frac{\left|\beta -2\right|}{\beta }\left({\left(\frac{{\left(z/c\right)}^{2}}{\left|\beta -2\right|}+1\right)}^{\alpha /2}-1\right) & \;\mathrm{otherwise}\; \end{array}$$

It can be seen that the adaptive loss function can be a variety of loss functions, such as the MSE, Cauchy, Charbonnier, and Welsch loss functions, by adjusting the value of the variable parameter β.

## 3. Proposed Model

### 3.1. Improved VMD

With a solid mathematical theoretical foundation, VMD can effectively separate the components of complex signals and greatly suppress mode mixing. However, the decomposition parameter of VMD is given in advance, which limits the performance of data decomposition. In order to overcome the shortcomings of VMD in a parameter setting, this paper proposes the incorporation of the decomposition method and MIC [42] to determine the most suitable number of decompositions *K*. The degree of decomposition is determined by calculating the MIC value between the original sequence y and the reconstructed sequence y′, and the MICyy′ value is positively correlated with the number of decomposition numbers *K*. The closer MICyy′ value is to 1, the less information is lost during VMD decomposition, indicating a more adequate decomposition.

### 3.2. IVMD-FE-Ad-Informer Model Framework

In consideration of the high volatility of wind power data, this paper introduces the improved VMD and FE methods to the Informer network with adaptive loss function, and the framework is shown in Figure 2. In the data processing stage, the original data are decomposed into *K* IMFs by IVMD. Next, FE is used to calculate the complexity of each IMFs, and the IMFs with similar values are reconstructed into new elements. In the model-building stage, the input variables for the model are obtained through feature selection using the MIC algorithm and then input into a robust Ad-Informer prediction model. In the results analysis stage, the wind power forecasting results are obtained by linearly superposing the predicted values of each element, followed by visualizing the forecasting curve.

### 3.3. Evaluation Indexes

The mean absolute error (MAE), root mean square error (RMSE), and coefficient of determination (R^2^) are used as evaluation indicators for the prediction performance of IVMD-FE-Informer and other benchmark models. The mathematical formula is as follows:(16)$$MAE=\frac{1}{N}\overset{N}{\sum }\left|q_{\mathrm{true}\;}(t)-q_{\mathrm{pred}\;}(t)\right|$$
(17)$$RMSE=\sqrt{\frac{1}{N}\sum_{t=1}^{N}{\left(q_{\mathrm{true}\;}(t)-q_{\mathrm{pred}\;}(t)\right)}^{2}}$$
(18)$$R^{2}=1-\frac{\sum_{t=1}^{N}{\left(q_{\mathrm{true}\;}(t)-q_{\mathrm{pred}\;}(t)\right)}^{2}}{\sum_{t=1}^{N}{\left(q_{\mathrm{true}\;}(t)-\bar{q}\right)}^{2}}$$
where $q_{\mathrm{true}\;}(t)$ and $q_{\mathrm{pred}\;}(t)$ denote the true and predicted values of wind power at time *t*, respectively, $\bar{q}$ is the mean value of $q_{\mathrm{true}\;}$, and *N* is the number of samples in the dataset.

## 4. Experiment and Analysis

In this section, four sets of experiments are conducted on datasets with different sampling intervals, capacities, and regions. Experiment 1 aims to describe the specific details of the data processing. Experiment 2 is designed as an ablation experiment to verify the prediction performance of the hybrid model. Experiment 3 mainly aims to design a comparative experiment to verify the viability and superiority of each module. Experiment 4 aims to verify the applicability and stability of the proposed model on different datasets. All experiments are run in Python 3.7 and Pytorch environment with Intel(R) Core (TM) i5-12500H CPU @ 4.50 GHz, 12 Cores, NVIDIA GeForce RTX 3050 GPU, a memory capacity of 16 Gb, and Windows 11 operating system.

### 4.1. Data Description

The experiments are mainly conducted on two complete datasets without missing values in this paper. Dataset A is based on a wind farm in Gansu, China, which was selected from 1 July to 30 September 2019, with a sampling interval of 15 min. Dataset A contains wind power, wind speeds at different heights (10 m, 30 m, 50 m, 70 m, and hub height), air temperature, air pressure, and humidity features. Dataset B was collected from the Sotavento Galicia wind farm in Spain from 18 January to 12 March 2020, with a sampling interval of 10 min. Dataset B contains only wind power, wind speed, and wind direction features. The wind power curves from different datasets are show in Figure 3.

The prediction process is the same for different datasets; in fact, dataset A is used for Experiments 1 to 3, and dataset B is used for Experiment 4. The datasets are divided into the training set, validation set, and test set in a ratio of 7:2:1, and the results of each experiment are obtained by taking the average of 10 iterations. The characteristics of the datasets, including number, maximum value (Max), minimum value (Min), mean, standard deviation (Std), and coefficient of variation (COV), are shown in Table 1.

### 4.2. Experiment 1: The Specific Details of Data Processing

The data processing part mainly includes data decomposition, new elements reconstruction, and feature selection, and in this part, the operation process and the selection of parameters for data processing will be specifically discussed.

#### 4.2.1. Data Decomposition

The IVMD algorithm solves the traditional VMD problem of *K* selection by calculating the MICyy′ value. The original wind power data are fed into the IVMD model, which is decomposed into *K* IMFs. Based on the results of MICyy′ corresponding to different values of *K* as indicated in Figure 4, it can be observed that the value of MICyy′ remains stable and constant for *K* = 16. The IMFs curve after IVMD decomposition and its corresponding spectrum diagram are shown in Figure 5. By observing the principal frequencies of different IMFs from Figure 5, it can be concluded that the IVMD algorithm proposed in this paper can effectively separate each IMF accurately.

#### 4.2.2. New Elements Reconstruction

The original wind power data are decomposed into 16 IMFs, and if all the sub-series are directly fed into the prediction model, it will increase the operational burden of the prediction model. Therefore, the complexity of these IMFs will be evaluated by FE, and then the IMFs with similar complexity will be reconstructed into new elements. After conducting extensive experiments, the values of m = 2 and r = 0.25std are found to be the optimal settings for achieving the best accuracy and running time of the model. The FE values of each IMF are shown in Figure 6, and the reconstructed new elements based on these FE values are shown in Table 2.

#### 4.2.3. Feature Selection

The computational efficiency and generalization ability of the model can be improved by removing some irrelevant or redundant features from the original dataset. Therefore, MIC is used to analyze the correlation between meteorological features and each element and extract typical features reflecting each element through MIC value. The confusion matrix of MIC is given in Figure 7. It can be found from Figure 7 that the influence characteristics of each element are different, reflecting the overall correlation and local characteristics, respectively. In order to select the features with the highest relevance to build the input variables, the MIC thresholds of each element are set to 0.5. The input feature selection results are shown in Table 3.

### 4.3. Experiment 2: Ablation Experiment

The purpose of conducting ablation experiments is to verify whether the complex hybrid model has improved the prediction accuracy as compared to simple combinatorial models and single models. The selected benchmark models are IVMD-FE-Ad-Informer, Ad-Informer, and Informer, of which the Informer model uses the MSE loss function. The parameters of AD-Informer are obtained using the grid search method, where the robustness parameter β is adaptively adjusted using the Adam optimizer. The specific parameters are shown in Table 4. The input size of the encoder and decoder is equal to the number of input variables of the model. The prediction curve results of each sub-mode after the training of the IVMD-FE-Ad-Informer model are shown in Figure 8, and the wind power prediction results can be obtained by superimposing them. The final forecasting curves of the ablation experiment are shown in Figure 9, and forecasting errors are shown in Table 5. Figure 9 not only portrays the overall trend of the test set but also amplifies the values from position 300 to 480 in order to offer an in-depth analysis of the predicted results. The main reason is that the wind power data within the test set from the 300th to the 480th position displays more sudden changes and a wider range of variation, thus providing a more comprehensive evaluation of the predictive performance of the proposed model.

From Figure 9, it can be seen that the Ad-Informer model is significantly closer to the true value than the Informer model at the inflection point, indicating that the proposed adaptive function can effectively mitigate the impact of errors at the abrupt points. Compared with the Ad-Informer model, the IVMD-FE-Ad-Informer model is closer to the real value, indicating that the data processing method can reduce the time delay in the process of prediction. According to Table 5, the Ad-Informer model requires less time than the Informer model, mainly attributed to the automatic adjustment of the adaptive loss function, which enables the model to obtain the optimal loss during the training process and enhance its robustness. The hybrid model proposed in this paper shows significant improvements over the AD-Informer and Informer models, with a decrease of 45.09% and 59.67% in MAE, 44.4% and 55.44% in RMSE, and an increase of 11.42% and 22.72% in R2, respectively. By comparing the considered models, it can be seen that IVMD-FE-Ad-Informer decomposes the original wind power data into finer granularity, which can better explore the internal features of wind power, resulting in a significant improvement in both prediction accuracy and performance.

### 4.4. Experiment 3: Comparative Experiment

To verify the superiority of each module, EMD-FE-Ad-Informer, IVMD-FE-Informer, IVMD-FE-LSTM, LSTM, and ANN are used as benchmark models in the comparison experiments, and the parameter settings of ANN and LSTM are the same as [19,43]. EMD decomposes wind power data into 11 IMFs by trail-and-error method, and then these IMFs are reconstructed into three new components (IMF1~IMF3, IMF4~IMF6, and IMF7~IMF11) using FE. The forecasting curves of different models are shown in Figure 10. The forecasting errors are shown in Table 6. The boxplots of the forecasting errors for each model are given in Figure 11.

Based on the results in Table 6, the IVMD-FE-Ad-Informer model outperforms the single prediction model and the other hybrid models across all evaluation metrics. It can be concluded that MAE decreased by about 35.68–60.32%, RMSE decreased by about 36.11–59.67%, and R^2^ increased by about 5.64–30.78%. According to Table 6 and Figure 10, it can be inferred that the IVMD algorithm has superior data decomposition ability compared to the traditional EMD algorithm under similar data processing. This improved ability enables the IVMD algorithm to more effectively reduce non-smooth features in the original data, resulting in smoother data and improved wind power prediction accuracy. Furthermore, the prediction accuracy of Ad-Informer is much higher than that of Informer and LSTM for the same data processing method, with R^2^ of 0.925, 0.889, and 0.808, respectively. While IVMD-FE-Ad-Informer is relatively time-consuming due to the implementation of the Ad-Informer prediction module five times after the IVMD-FE data preprocessing, it demonstrates a closer resemblance to the actual curve and produces the smallest forecasting errors. It can be indicated that the model proposed in this paper is an optimal combined model with high prediction performance.

### 4.5. Experiment 4: The Stability of IVMD-FE-Ad-Informer Forecasting

The experimental results demonstrate that IVMD-FE-Ad-Informer outperforms other benchmark models on dataset A and exhibits considerable wind power prediction ability. However, the statistical distributions of wind power data vary across different time intervals, regions, and capacities, which may lead to the phenomenon of unstable forecasting. Therefore, the stability and applicability of the model still need further discussion. In this section, EMD-FE-Ad-Informer, Ad-Informer, LSTM, and ANN are used as benchmark models on dataset B, which are collected from the Sotavento Galicia wind farm in Spain at 10 min sampling intervals. The parameter-setting method of this experiment is the same as Experiment 3, and the specific parameter settings of each algorithm are shown in Table A1, Appendix A. The forecasting curves of dataset B are shown in Figure 12, and the forecasting errors are shown in Table 7.

According to Figure 12 and Table 7, the results obtained from dataset B are comparable to those from dataset A, indicating that the model proposed in this paper has high stability and generalization ability on different datasets. From Figure 12, IVMD-FE-Ad-Informer exhibited the closest fit to the true values among all the considered models, with the EMD-FE-Ad-Informer following closely behind. It can also be seen from Table 7 that the IVMD-FE-Ad-Informer has the best prediction performance with regard to MAE, RMSE, and R^2^, which are 83.01 kW, 60.43 kW, and 0.962, respectively. The results further confirm that the IVMD algorithm is a superior and effective method for wind power data decomposition.

Based on the experimental results of the two different datasets, it is apparent that the IVMD-FE-Ad-Informer outperforms other benchmark models in terms of all evaluation metrics and has the closest fit of prediction curves to the true values. Meanwhile, the COV value is introduced for further analysis of the influence of prediction accuracy on different datasets. This value is a typical indicator of the degree of data fluctuation, with more volatile data having a higher COV value [44]. It also can be concluded that the accuracy of the proposed model prediction is inversely related to the degree of fluctuation in the original data. For example, when using Ad-Informer to forecast wind power on dataset A, the R^2^ is 0.866, whereas, on dataset B with a higher COV, R^2^ is slightly lower at 0.858. Furthermore, the superiority of the proposed model in terms of prediction performance becomes more prominent as the original wind power sequence contains more nonlinear features. The outstanding contribution is the development of an adaptive loss function, which can accurately identify and predict violent changes in wind power, thereby effectively mitigating the impact of outliers.

## 5. Conclusions

The actual operation of wind farms is influenced by various factors such as weather conditions, season variation, and atmospheric circulation, which can lead to numerous outliers and non-smooth features in the wind power data. The presence of such factors brings many obstacles to achieving the further improvement of accuracy and performance of wind power prediction. Thus, an adaptive hybrid model for wind power prediction based on improved VMD, FE, and Informer in conjunction with adaptive loss function is proposed in this paper. The IVMD-FE-Ad-Informer model is a promising hybrid model that enables adaptive forecasting of stochastically fluctuating wind power data, and its main advantages are summarized as follows:The IVMD-FE-Ad-Informer is a hybrid model that demonstrates high accuracy and better robustness by integrating the advantages of multiple technologies, outperforming the basic EMD- FE-Ad-Informer, Ad-Informer, LSTM, and ANN. The results of the proposed model obtained from the Spanish and Chinese datasets demonstrate a significant improvement compared to benchmark models, with a maximum reduction of 57.89% in MAE, 57.03% in RMSE, and a maximum increase of 30.78% in R^2^;Compared with traditional data decomposition methods, VMD improved by MIC can better mine the nonlinear features of the original data, which effectively improves the data quality and reduces the difficulty of prediction;Based on a comprehensive analysis of experimental results, the adaptive loss function has a rapid response to non-Gaussian distributed wind power data, which can react quickly to outliers and predict variation trends;By prediction experiments on wind farm datasets with different sampling intervals, capacities, and regions, the proposed model shows the best prediction results and closest proximity to the true value. It can be demonstrated that IVMD-FE-Ad-Informer has remarkable generalization ability and broad prospects in wind power prediction.

As can be seen from the above, the hybrid wind power prediction model that combines the advantages of several algorithms has higher prediction accuracy and better robustness. However, there are some problems in this study that need to be improved in the future. Firstly, in this paper, only the correlation factor is considered in feature selection, while F-score and sensitivity factors are not taken into account. In future work, the analysis of the relationship between other variables and wind power using F-score and sensitivity will be conducted to reduce the redundancy of massive data. Then, the parameter selection in this paper may not be precise enough, and to address this issue, the optimization algorithm will be introduced to overcome the sensitive defect of deep-learning network-parameter selection.

## Figures and Tables

**Figure 1 entropy-25-00647-f001:**
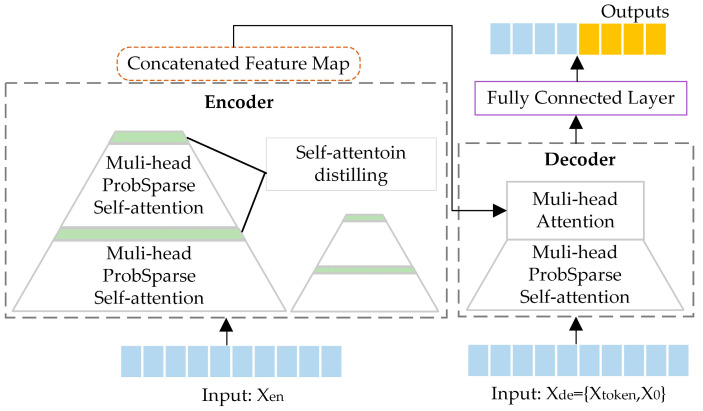
The structure of the Informer.

**Figure 2 entropy-25-00647-f002:**
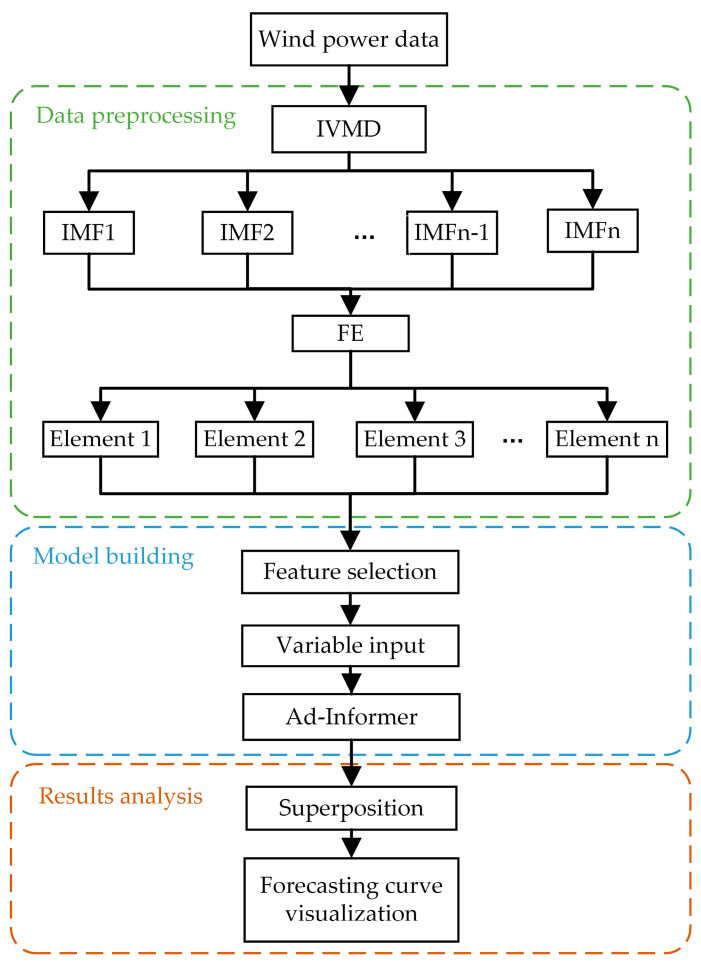
The framework of IVMD-FE-Ad-Informer model.

**Figure 3 entropy-25-00647-f003:**
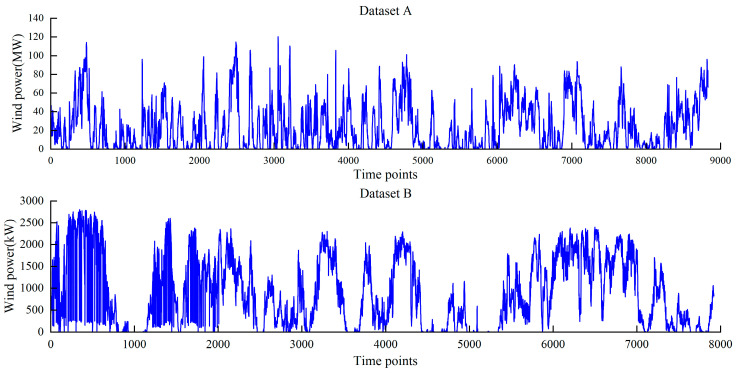
The curve of datasets.

**Figure 4 entropy-25-00647-f004:**
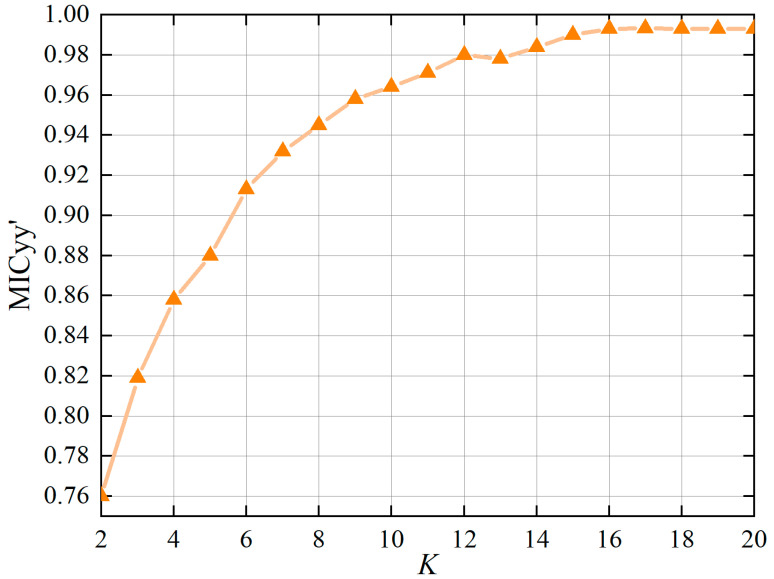
The curve of MICyy′ vs. *K*.

**Figure 5 entropy-25-00647-f005:**
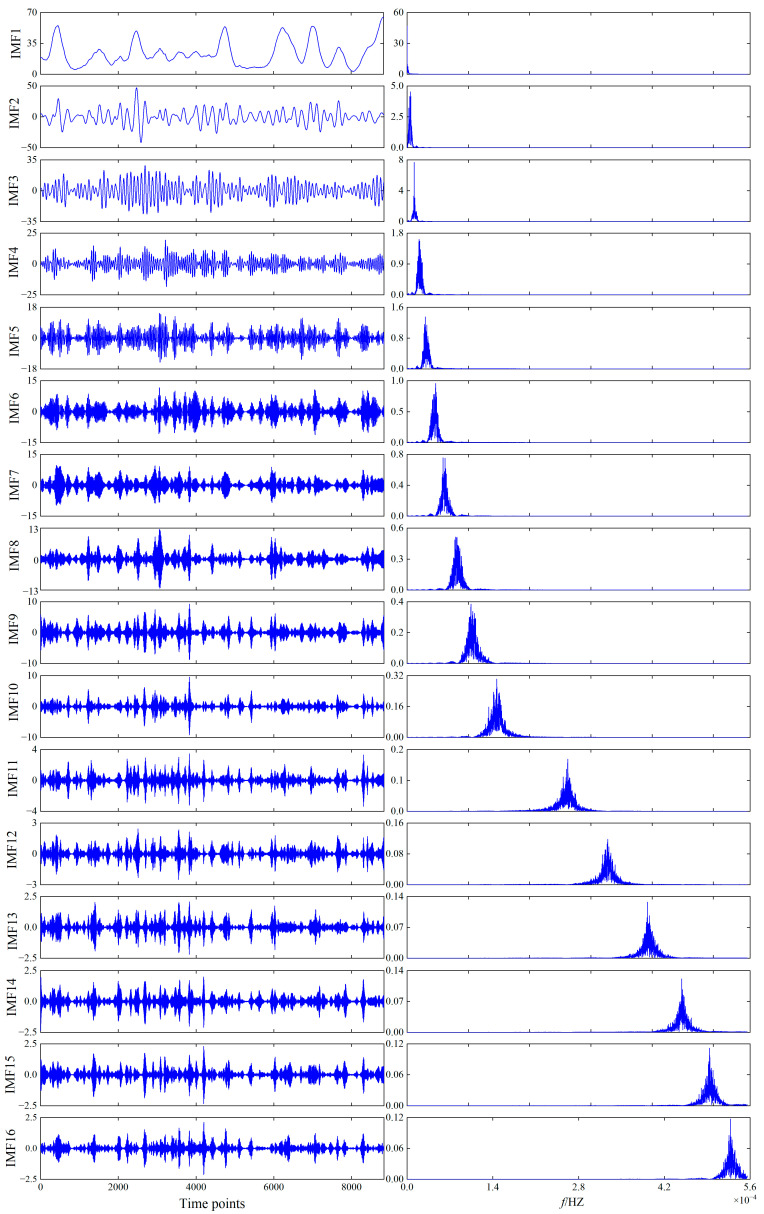
The results of the IVMD algorithm. The IMFs curves are shown (**left**), and the spectral densities corresponding to the IMFs are shown (**right**).

**Figure 6 entropy-25-00647-f006:**
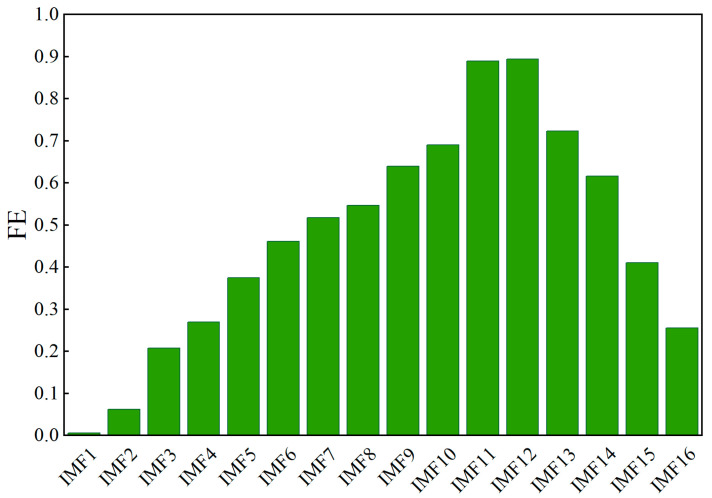
The FE value of each IMF.

**Figure 7 entropy-25-00647-f007:**
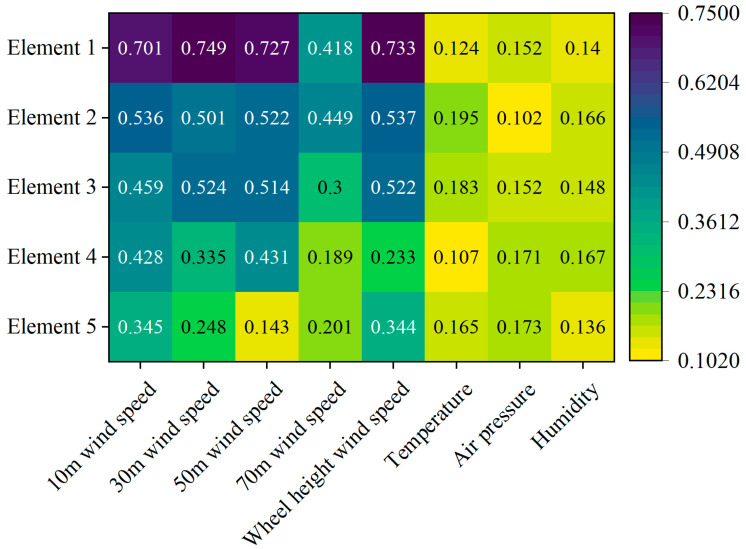
The confusion matrix of MIC.

**Figure 8 entropy-25-00647-f008:**
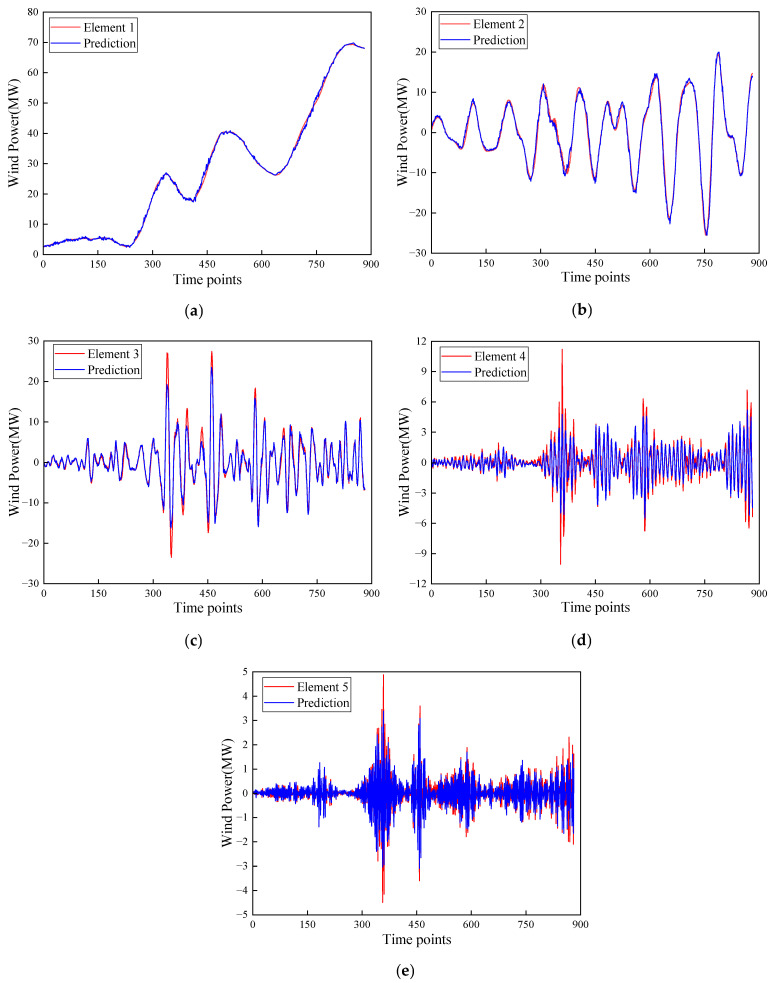
The prediction curve results of each sub-mode. (**a**) Element 1 prediction curve; (**b**) Element 2 prediction curve; (**c**) Element 3 prediction curve; (**d**) Element 4 prediction curve; (**e**) Element 5 prediction curve.

**Figure 9 entropy-25-00647-f009:**
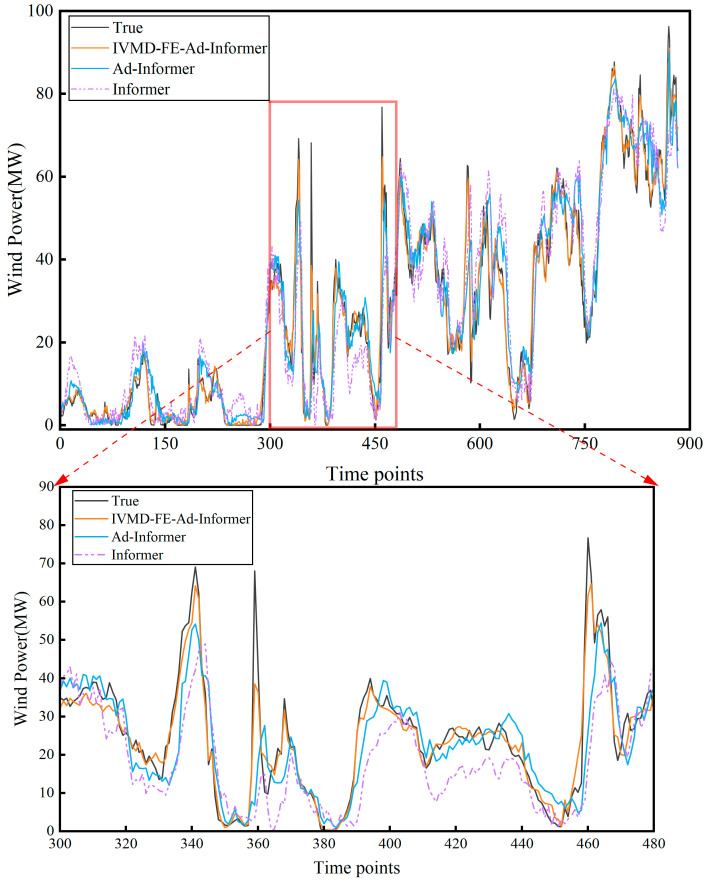
The forecasting curves of the ablation experiment. The overall forecasting trends are shown at the (**top**), and the local enlargement is shown at the (**bottom**).

**Figure 10 entropy-25-00647-f010:**
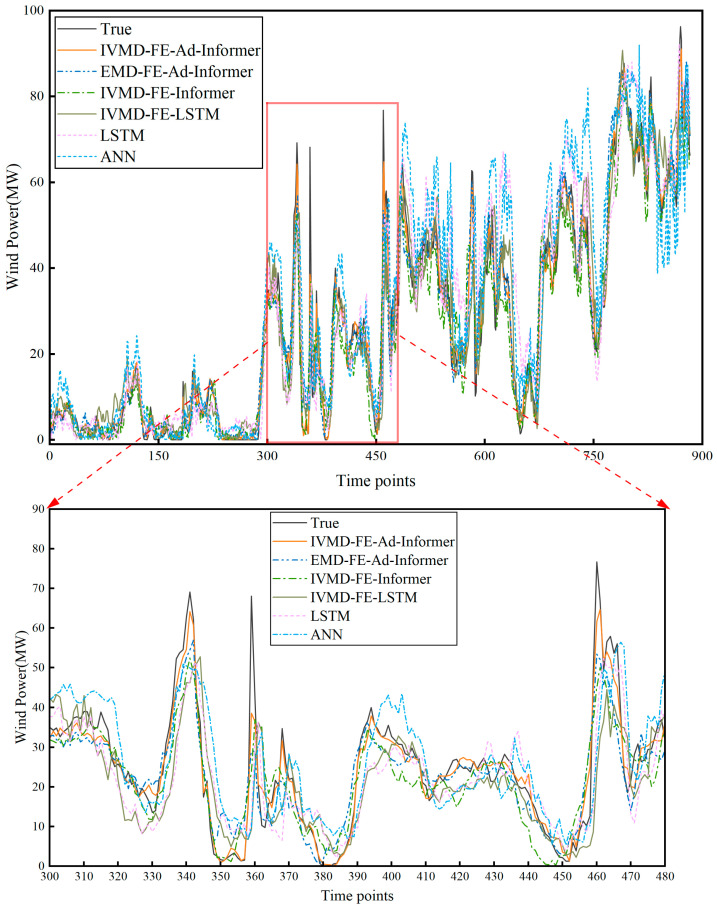
The forecasting curves of different models. The overall forecasting trends are shown at the (**top**), and the local enlargement is shown at the (**bottom**).

**Figure 11 entropy-25-00647-f011:**
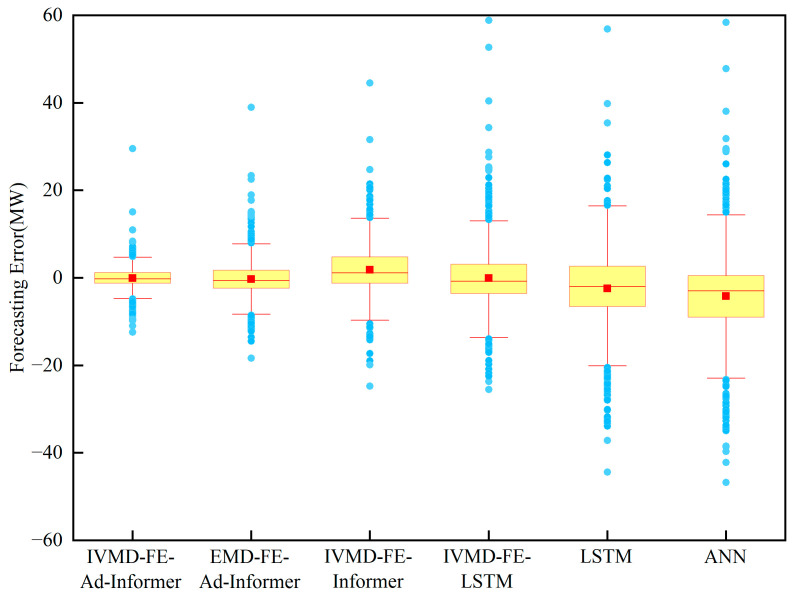
The boxplots of different models.

**Figure 12 entropy-25-00647-f012:**
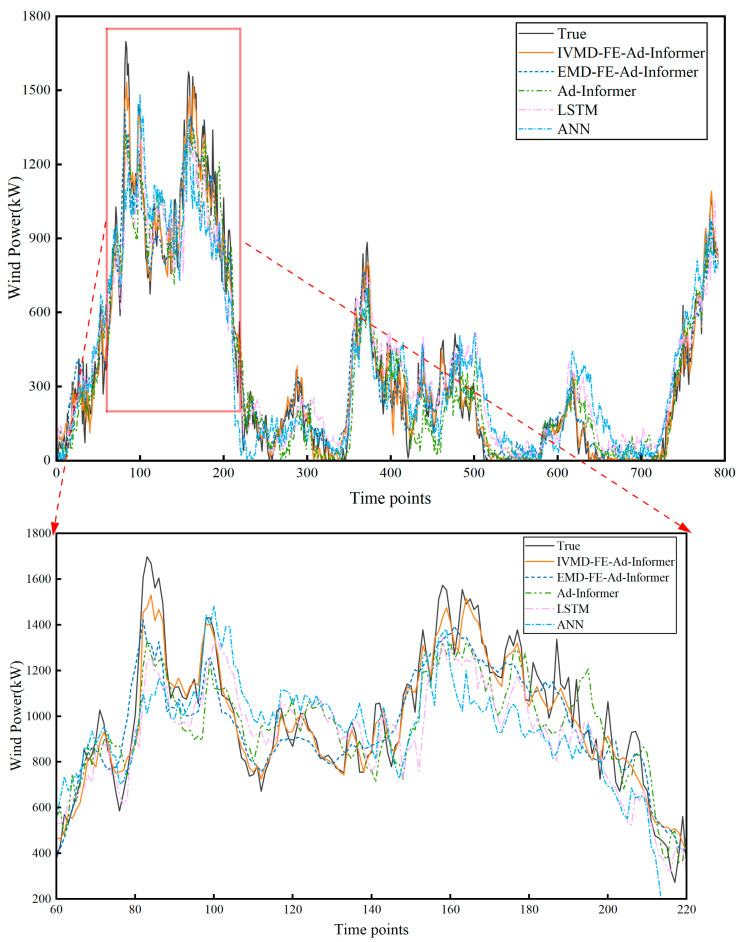
The forecasting curves of different datasets. The overall forecasting trends are shown at the (**top**), and the local enlargement is shown at the (**bottom**).

**Table 1 entropy-25-00647-t001:** The characteristics of datasets.

Dataset	Number	Max (MW)	Min (MW)	Mean (MW)	Std (MW)	COV
Dataset A	8832	120.43	0	23.51	24.74	1.0523
Dataset B	7920	2.803	0	0.8976	0.7885	0.8784

**Table 2 entropy-25-00647-t002:** New elements reconstruction.

Reconstruction Elements	IMFs
Element 1	IMF1, IMF2
Element 2	IMF3, IMF4, IMF16
Element 3	IMF5, IMF6, IMF7, IMF8, IMF15
Element 4	IMF9, IMF10, IMF13, IMF14
Element 5	IMF11, IMF12

**Table 3 entropy-25-00647-t003:** The feature selection results.

Element	Input Variables
Element 1	Element 1, 10 m, 30 m, 50 m, wheel height wind speed
Element 2	Element 2, 10 m, 50 m, wheel height wind speed
Element 3	Element 3, 30 m, 50 m, wheel height wind speed
Element 4	Element 4
Element 5	Element 5

**Table 4 entropy-25-00647-t004:** Parameter setting of the Ad-Informer.

Parameters	Values
Input sequence length	96
Start token length	24–96
Prediction sequence length	24–96
Num of encoder layers	3
Num of decoder layers	2
Input size of encoder	5-1
Input size of decoder	5-1
Decoder output	1
Num of heads	8
Dimension of model	512
Probsparse attention factor	5
Early stopping patience	5
Learning rate	0.0001
Dropout	0.05
Epochs	100
Scale factor	1.2
Optimizer	Adam
Gpu	Cuda0

**Table 5 entropy-25-00647-t005:** Forecasting errors of ablation experiment.

Model	MAE (MW)	RMSE (MW)	R^2^	Time (s)
IVMD-FE-Ad-Informer	3.19	4.67	0.956	1633.21
Ad-Informer	5.81	8.40	0.858	177.13
Informer	7.91	10.48	0.779	165.559

**Table 6 entropy-25-00647-t006:** Forecasting errors of different models.

Model	MAE (MW)	RMSE (MW)	R^2^	Time (s)
IVMD-FE-Ad-Informer	3.19	4.67	0.956	1633.21
EMD-FE-Ad-Informer	4.96	7.31	0.905	1362.47
IVMD-FE-Informer	5.81	8.40	0.889	1878.63
IVMD-FE-LSTM	6.63	9.79	0.808	1732.57
LSTM	7.86	10.95	0.759	305.11
ANN	8.04	11.58	0.731	81.02

**Table 7 entropy-25-00647-t007:** Forecasting errors of different datasets.

Model	MAE (kW)	RMSE (kW)	R^2^	Time (s)
IVMD-FE-Ad-Informer	83.01	60.43	0.962	1076.34
EMD-FE-Ad-Informer	115.89	70.75	0.914	671.47
Ad-Informer	144.51	105.46	0.866	156.91
LSTM	186.63	131.04	0.762	228.88
ANN	197.16	140.62	0.746	62.15

## Data Availability

The data used to support the findings of this study are available from the corresponding author upon request.

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
