# Peer review of "An Adaptive Hybrid Model for Wind Power Prediction Based on the IVMD-FE-Ad-Informer"

_entropy, 2023, doi:10.3390/e25040647_

Round 1

Reviewer 1 Report

In this paper the authors have proposed a new prediction model for wind power which is based on the Informer with IVMD and FE. The model have shown significant improvement of prediction of wind power in terms of accuracy. I enjoy reading this paper and recommend publication.

However, there is a minor concern about the time to run the model for prediction. From Table 6 and 7, it seems that the new model is a lot more time consuming compared to other models. Could the authors comment that?

Technical comments:

Title: The superscript for the authors should be the same from the same institution.

L228: python→Python, and I think you mean 12600K and RTX3050

Figure 8, 9 and 10: Please make the title clearer in terms of why and how you choose the section to amplify.

Author Response

Response to Reviewer 1 Comments

Dear Editor and Reviewers,

Thanks very much for taking your time to review this manuscript. I really appreciate all your comments and suggestions! Please find my itemized responses in below and my revisions in the re-submitted files. Thanks again!

Point 1: However, there is a minor concern about the time to run the model for prediction. From Table 6 and 7, it seems that the new model is a lot more time consuming compared to other models. Could the authors comment that?

Response 1: We thank the reviewer for pointing out this issue. We acknowledge that IVMD-FE-Ad-Informer is relatively time-consuming, but it demonstrates a closer resemblance to the actual curve and produces the smallest forecasting errors. Taking the result analysis in Table 6 as an example, after data processing of IVMD-FE, the prediction module has been implemented for 5 times. However, after data processing for EMD-FE, the prediction module has been implemented for 3 times, while no data processing is required for a single model, and the prediction module has been only implemented for 1 time. The same reason in Table 7 causes the new model to be relatively time-consuming. In the revised manuscript, we provide a detailed explanation and description this issue in Section 4.3 (lines 371-375).

Point 2: Title: The superscript for the authors should be the same from the same institution. L228: python→Python, and I think you mean 12600K and RTX3050

Response 2: Thank you so much for your careful check. We have updated the author superscripts to ensure consistency for authors from the same institution. Additionally, we have made the necessary corrections in revised manuscript.

Point 3: Figure 8, 9 and 10: Please make the title clearer in terms of why and how you choose the section to amplify.

Response 3: Thank you for pointing out this problem in our manuscript. According to the revised manuscript, we have revised in the titles of Figure 9, 10, and 11 (previously labeled as Figure 8, 9, and 10 in the original manuscript). Specifically, we have added a brief description which graph displays the overall trends and which graph displays a local enlarged view.

Furthermore, we have also explained in the revised manuscript (Section 4.3, lines 321-326) that we selected the local amplification area based on the fact that the wind power data exhibits more sudden changes and a wider range of variation, making it more suitable for a comprehensive evaluation of our model's predictive performance.

Reviewer 2 Report

The paper presents an adaptive hybrid model for deep learning wind power prediction based on improved variation mode decomposition and fuzzy entropy and informer combination with adaptive loss function. The following comments are for the authors of the paper.  

1. The paper has some grammatical errors making it difficult to comprehend some portions of the paper.

2. In Fig. 2, how are the various modes (mode 1, mode 2, mode 3,..) operated? What are the features been extracted, the variable input and the conditions for superposition.

3. Is the MIC matrix was presented in Fig. 6 similar to the confusion matrix topology?

4. Apart from the wind power variable, other variables of the study should be considered based on the F-score, sensitivity, selectivity of metrics. 

5. What are the impacts of the other components of wind speed like the base, gust, ramp etc speed components on the presented results.  

Reviewer 3 Report

The problem is interesting. However, some improvements must be made.

(a) Plz keep the abstract concise with useful information.

(b) Your literature review must be improved. Some old studies should be deleted. The wind speed forecasting is also closely related to your topic. The literature review should discuss more related forecasting studies to highlight the research importance (Explainable temporal dependence in multi-step wind power forecast via decomposition based chain echo state networks; Multivariate wind speed forecasting based on multi-objective feature selection approach and hybrid deep learning model; Forecasting energy consumption and wind power generation using deep echo state network; Short-Term Prediction of 80-88 km Wind Speed in Near Space Based on VMD-PSO-LSTM; etc).

(c) How to set the related parameters in Table A1 for better performance?

(d) The conclusion part is weak and needs to be enriched. A discussion of the advantage and underlying drawbacks of the proposed method is welcome in the conclusion part.

(e) Try to analyze Fig.11 from the theory aspect.

Reviewer 4 Report

The authors proposed an adaptive IVMD-FE-Ad-Informer hybrid model for wind power prediction. Experimental data was used to verify the stability and effectiveness of the model. The predicted performance and accuracy are enough reliable. In general, the paper and its research and results are relevant, have an applied side, and provide guidance for future work.

Comments for improving the article:

Abstract should be slightly shorter: max of 200 words.

Closed form of references citation is not recommended (line 43, line 51).

The authors have done a good overview, which concludes with a presentation of what the paper itself goes on to say, and what scientific gaps are being addressed. However, the question arises as to what was the main objective of the study, whether it was the development of the model itself, the improvement of the method or some other objective.

Figure 6 is only presented. The results obtained should be discussed.

Section 4.2.4 could be merged with another section as it presents only one table of parameters.

English should be revised.

As a technical scientific paper, I would recommend some numerical indicators in the conclusions that relate to your proposed model and the results it produces compared to other methods.

Round 2

Reviewer 3 Report

The revision is satisfactory.